# Evaluation of Respiratory Resistance as a Predictor for Oral Appliance Treatment Response in Obstructive Sleep Apnea: A Pilot Study

**DOI:** 10.3390/jcm10061255

**Published:** 2021-03-18

**Authors:** Hiroyuki Ishiyama, Masayuki Hideshima, Shusuke Inukai, Meiyo Tamaoka, Akira Nishiyama, Yasunari Miyazaki

**Affiliations:** 1Dental Anesthesiology and Orofacial Pain Management, Graduate School of Medical and Dental Sciences, Tokyo Medical and Dental University, Tokyo 113-8549, Japan; h.ishiyama.rpro@tmd.ac.jp (H.I.); anishi.tmj@tmd.ac.jp (A.N.); 2Dental Clinic for Sleep Disorders (Apnea and Snoring), Oral and Maxillofacial Rehabilitation, Dental Hospital, Tokyo Medical and Dental University, Tokyo 113-8549, Japan; 3Removable Partial Prosthodontics, Oral Health Sciences, Graduate School of Medical and Dental Sciences, Tokyo Medical and Dental University, Tokyo 113-8549, Japan; inurpro@tmd.ac.jp; 4Department of Respiratory Physiology and Sleep Medicine, Graduate School of Medical and Dental Sciences, Tokyo Medical and Dental University, Tokyo 113-8549, Japan; meiyou2.pulm@tmd.ac.jp; 5Department of Respiratory Medicine, Graduate School of Medical and Dental Sciences, Tokyo Medical and Dental University, Tokyo 113-8549, Japan; miyazaki.pilm@tmd.ac.jp

**Keywords:** obstructive sleep apnea, oral appliance, predictor, respiratory resistance, impulse oscillometry

## Abstract

The aim of this study was to determine the utility of respiratory resistance as a predictor of oral appliance (OA) response in obstructive sleep apnea (OSA). Twenty-seven patients with OSA (mean respiratory event index (REI): 17.5 ± 6.5 events/h) were recruited. At baseline, the respiratory resistance (R20) was measured by impulse oscillometry (IOS) with a fitted nasal mask in the supine position, and cephalometric radiographs were obtained to analyze the pharyngeal airway space (SPAS: superior posterior airway space, MAS: middle airway space, IAS: inferior airway space). The R20 and radiographs after the OA treatment were evaluated, and the changes from the baseline were analyzed. A sleep test with OA was carried out using a portable device. The subjects were divided into Responders and Non-responders based on an REI improvement ≥ 50% from the baseline, or REI < 5 after treatment, and the R20 reduction rate between the two groups were compared. The subjects comprised 20 responders and 7 non-responders. The R20 reduction rate with OA in responders was significantly greater than it was in non-responders (14.4 ± 7.9 % versus 2.4 ± 9.8 %, *p* < 0.05). In responders, SPAS, MAS, and IAS were significantly widened and R20 was significantly decreased with OA (*p* < 0.05). There was no significant difference in non-responders (*p >* 0.05). A logistic multiple regression analysis showed that the R20 reduction rate was predictive for OA treatment responses (2% incremental odds ratio (OR), 24.5; 95% CI, 21.5–28.0; *p* = 0.018). This pilot study confirmed that respiratory resistance may have significant clinical utility in predicting OA treatment responses.

## 1. Introduction

Obstructive sleep apnea (OSA) is characterized by intermittent obstruction of the upper airway during sleep and by repeated apnea and hypopnea, which cause intermittent hypoxia and sleep fragmentation, reducing sleep quality [1,2,3]. Oral appliance (OA) therapy is a treatment option for OSA. An OA widens the upper airway by advancing the mandible; this prevents upper airway obstructions during sleep [4,5]. Although its efficacy is inferior to that of continuous positive airway pressure (CPAP) therapy, OA therapy has high compliance [6,7] and not only improves apnea, hypopnea, and symptoms of OSA, but has recently also been shown to be effective on the cardiovascular comorbidity in OSA, such as hypertension and arrhythmia [8]. The success rate of OA therapy in patients with mild–severe OSA was reported to be 35–64% in a recent review [9] and individual variability in response to OA treatment represented a significant clinical challenge for implementing this therapy [10]. According to the clinical practice guidelines for the treatment of OSA, OA therapy is recommended mainly for patients with mild to moderate OSA, or patients who cannot tolerate CPAP [11]. Therefore, the indication of OA in current OSA treatment is determined based on the severity of the Apnea Hypopnea Index (AHI). However, OA therapy is reported to be effective even in severe cases [12,13], and thus, severity by itself is not considered to accurately indicate the need for OA therapy.

To determine whether it is indicated, it is also necessary to conduct appropriate pretreatment evaluations in addition to the AHI, in order to predict the potential treatment effects. To determine OA therapy indications, evaluating the relationship of morphological changes of the upper airway and an obstruction site during mandibular advancement is highly important. The factors that are affected by changes in the upper airway include breathing sounds, respiratory resistance, and muscular function. However, a method for predicting the treatment effects of OA in a simple and non-invasive procedure using these parameters has not yet been established.

The Impulse Oscillation System (IOS) is a method that assesses breath dynamics while breathing at rest [14], unlike conventional pulmonary function testing (spirometry) that requires forced breathing. The air pressure oscillations are inserted into the oral cavity through a mouthpiece attached to a device, in order to measure respiratory resistance from the differences in the phases between the pressure and airflow, which change depending on the diameter of the airway and its elasticity. The localization of resistance can be inferred using frequency characteristics, and IOSs have been used for the diagnosis and treatment of bronchial asthma and chronic obstructive pulmonary disease (COPD) [15]. Its advantages include the fact that testing can be conducted in a short period of time and that the measuring method is non-invasive and easy to perform. The aim of this study was to measure respiratory resistance using the IOS in the waking state and to examine how the changes in respiratory resistance resulting from mandibular advancement reflect the treatment effects of OA.

## 2. Materials and Methods

### 2.1. Subjects

The subjects were adult patients, diagnosed with OSA (AHI ≥ 5), and recruited from the Dental Clinic for Sleep Disorders (Apnea and Snoring) from Tokyo Medical and Dental University Dental Hospital. The following patients were excluded: (1) those with fewer than 19 remaining teeth, including the residual anterior teeth; (2) those receiving combined OA and CPAP therapy. (3) Those regularly using a sleep-inducing drug; (4) those with a mandibular advancement <9 mm; (5) those having a medical history of respiratory disease or suffering from a respiratory disease; (6) those allergic to steroids and vasoconstrictors; (7) those with mental illness; (8) those having temporomandibular disorders with pain or trismus; and (9) those having any caries or periodontal diseases that required treatment. All subjects were provided verbal and written information about the study and signed informed consent forms. The study protocol was approved by the Tokyo Medical and Dental University Dental Hospital ethics committee (protocol code D2012-024, 9 March 2016). The study was registered at UMIN-CTR (https://www.umin.ac.jp/ctr/ (accessed on 28 August 2014); UMIN000014984; Development of treatment effect prediction method of oral appliance therapy for obstructive sleep apnea syndrome).

### 2.2. Sleep Test

In order to assess the efficacy of the OA treatment, an out-of-center sleep test (OCST) was performed before and after the treatment. During testing, we used a Type-3 portable sleep testing device (Pulsleep LS-120S; Fukuda Denshi, Tokyo, Japan), which is capable of measuring the SpO2, pulse rate, airflow, snoring, and posture [16]. Expiratory and inspiratory flow was measured using a nasal cannula, and respiratory conditions during sleep were recorded. The SpO2 and pulse rate were confirmed using a fingertip pulse oximeter.

The sleep test was performed for three consecutive nights, and mean values were used as representative values. When the test data were inconclusive, they were excluded from the analysis. After confirming the original waveforms, data were manually analyzed, and a standardized procedure was applied to score the results [17]. The data were all scored by a single researcher (H.I.), who was blinded to the participants’ IOS data. The subjects were asked to keep a sleep diary and to record their sleep duration on the days of testing (including bedtime and the time they woke up), as well as the number and times of nocturnal awakenings, which were used as references for the analyses.

For analysis, the following values of the sleep test results were assessed: the REI (Respiratory Event Index) and the lowest SpO2. REI indicates the number of instances of apnea and hypopnea per hour of measurement. Apnea is defined as a 90% reduction in airflow for at least 10 seconds, and hypopnea is defined as ≥30% reduction in airflow for at least 10 seconds, associated with ≥3% reduction in oxygen saturation [17]. OSA was defined as a REI ≥ 5, and classified as mild (REI 5.0–14.9), moderate (REI 15.0–29.9), and severe (REI ≥ 30) [18].

### 2.3. Oral Appliance

A custom-made, monobloc, mandibular advancement oral appliance, made from a 2.0-mm polyethylene plate (Erkodur, Erkodent Inc., Pfalzgrafenweiler, Germany), was prescribed for all participants. The absolute range of the maximum mandibular advancement was measured using the George Gauge (Great Lakes Orthodontics, Ltd., New York, NY, USA) [19]. The amount of mandibular advancement in this study was set at 50–70% of the maximum, considering the discomfort and pain of the temporomandibular joint and masticatory muscles [20].

During follow-up of 2 months after the OA provision, the changes in OSA symptoms and the degrees of side effects that were associated with OA use were evaluated. The appliance was incrementally titrated to either a maximal comfortable protruded position of the mandible or a resolution of snoring and daytime symptoms [21]. Increased advancement of the appliance was facilitated by the separation of the upper and lower components of the appliance, and then repositioning at a more advanced mandibular position.

### 2.4. Assessment of Treatment Outcome

After the OA was fully adjusted (approximately 2.5 months after baseline), the OCST was performed with the OA in place. In this study, the efficacy of the OA treatment was evaluated using the relative REI improvement rates. The REI improvement rate was calculated using the following formula: [(REI before treatment) – (REI after treatment)]/(REI before treatment) × 100. Subjects with a rate of REI improvement ≥ 50%, or REI < 5 after treatment, were defined as responders, and those with a rate < 50% were defined as non-responders [22]

### 2.5. IOS and Test Task

Master screen IOS-J (Jaeger, Wurzburg, Germany) was used to assess respiratory resistance (Figure 1). This device releases impulse-like acoustic signals (frequency: 0–35 Hz) from a round speaker to detect the intraoral pressure and flow rate during breathing at rest, and then conducts Fourier transformations of the data to analyze respiratory resistance. IOS parameters, such as resistance at 5 Hz (R5) and resistance at 20 Hz (R20), were recorded. R5 and R20 represent the total airway resistance and resistance of the region from the upper airway through to the central airway, respectively (Figure 2) [14]. R20 was used for the analysis in this study, due to its usefulness in the prediction of AHI values [23].

A single researcher (S.I.) measured the patients’ respiratory resistances in the waking state. The researcher was blinded to the results of sleep test. The subjects were in the supine position and put on a nasal mask without any air leaks during the measurements. Because the measurements took place via the nasal cavity, the congestion in the nasal mucosa tended to occur over time, and accurate measurements of respiratory resistances can be hampered [24]. To prevent this, subjects were instructed to spray a vasoconstrictor nasal drop (cor-tyzine nasal solution; Tetrahydrozoline hydrochloride-prednisolone) into the nasal cavity several times, and measurements were performed 10 minutes later [25].

At the time of measurement, the subject was instructed to breathe while at rest. Respiratory resistance was measured three times under each condition with and without OA, and the mean values were used as the representative values. The measurements without OA were performed in the intercuspal position.

### 2.6. Cephalogram

In all recruited patients, lateral cephalometric radiographs were obtained in order to evaluate the subjects’ maxillofacial morphology at baseline, as well as the changes in the pharyngeal airway space with and without OA. The morphology of the pharyngeal airway on the lateral cephalogram is known to be largely affected by the skeleton, body type, posture, and respiratory phase; the comparative reproducibility of the upper airway morphology cannot be considered to be high [26]. Therefore, the subjects were placed in an upright position with the Frankfurt plane parallel to the floor, and photographed in the end-tidal position [27]. A cephalometric analysis was carried out as previously described (Figure 3) [28]. An analysis was performed by a single researcher (H.I.), and the names of patients were blinded.

### 2.7. Statistical Analysis

Continuous variables were described as mean ± SD for variables with a normal distribution and median (interquartile range) for variables with a non-normal distribution. Normality of distribution was assessed using the Shapiro-Wilk test.

The comparisons between groups were made using Student’s t-test for variables with a normal distribution and a Mann-Whitney U test for variables with a non-normal distribution, and Fisher’s exact test was used for categorical variables. The within-subject comparisons in each group were made using pared t-test for variables with a normal distribution and Wilcoxon signed-rank test with a non-normal distribution. The effect size (Cohen’s d) of the continuous variables was calculated. The R20 reduction rate with OA was calculated using the formula: [(pre-treatment R20) − (post-treatment R20]/(pre-treatment R20) × 100. In order to determine the predictive factors for the efficacy of the OA treatment, a binominal logistic regression (stepwise method) was performed. In the analysis, the explanatory variables were set as the responders and non-responders in the OA treatment, and the objective variables were set as the R20 reduction rates and the conventional predictors (sex, age, BMI, baseline-REI, and MR-H) [29,30,31]. In addition, to obtain a value that allows the highest discriminability for the prediction of the efficacy of OA therapy, the receiver operating characteristic curve and the area under the curve (AUC) were calculated in order to determine the significant factor in the regression analysis, and the cut-off value was determined using the Youden index. An odds ratio (OR) ≥2 or ≤0.5 was considered a clinically meaningful predictor. A value of *p* < 0.05 was considered statistically significant. All statistical analyses were performed using the SPSS version 21.0 software (IBM, Inc., Armonk, NY, USA).

## 3. Results

### 3.1. Baseline Characteristics of the Subjects

Thirty-five OSA patients were recruited into this study, and eight patients refused to participate in the study. Twenty-seven patients, with mild (*n* = 9), moderate (*n* = 17), and severe (*n* = 1) OSA, completed the full study protocol. The subject characteristics at baseline are shown in Table 1. The majority (81.5%) of patients were men. The median age of all subjects was 65.0 (56.0–70.0) years; their median Body Mass Index (BMI) was 25.1 (22.2–26.8) kg/m^2^. The mean REI of the subjects was 17.5 ± 6.5 events/h. The median Epworth Sleepiness Scale (ESS) of the subjects was 8.0 (6.0–11.0). Of the 27 subjects, 20 subjects were responders and 7 subjects were non-responders. There was a significant difference due to sex at baseline between responders and non-responders, whereas there was no significant difference in age, BMI, ESS, and R20. In addition, there was no significant difference in the ratio of the mandibular advancement after the OA adjustment, nor in any parameters representing maxillofacial morphology (SNA, SNB, ANB, PNS-P, or MP-H) between responders and non-responders. Furthermore, there was no significant difference in the pharyngeal airway space (SPAS, MAS, and IAS) on the cephalogram between the two groups.

### 3.2. The Comparison of the Responders and the Non-Responders in OA Therapy

The values of each parameter with and without the OA are shown in Table 2. With respect to within-subject comparisons, the responders showed a significant decrease in REI, R20, and ESS (*p* < 0.01) and a significant increase in LowestSpO2 with OA (*p* < 0.05). In the cephalogram, the responders showed a significant increase in SPAS, MAS, and IAS with OA (*p* < 0.05). In non-responders, the REI was significantly decreased with OA (*p* < 0.05), whereas there were no significant differences in the other parameters (*p* > 0.05). The results from the R20 reduction rate comparison between responders and non-responders are shown in Figure 4. The R20 reduction rate was significantly higher in responders compared to non-responders (*p* < 0.01; Cohen’s d = 1.46).

### 3.3. The Predictors Associated with Oral Appliance Treatment Success

The results of the binary logistic regression are shown in Table 3. The R20 reduction rate was shown to be an independent predictor of OA treatment response (2% incremental OR, 24.5; 95% CI, 21.5–28.0). However, sex, age, BMI, baseline-REI, and MP-H were not significant predictors. The ORs of the R20 improvement rates are presented for 0.1% increments, 1% increments, and 2% increments. The calculated cut-off value of the R20 reduction rate from the Youden index was 8.6% (AUC: 0.839, sensitivity: 0.80, specificity: 0.86) (Figure 5).

## 4. Discussion

This study demonstrated that the higher the respiratory resistance reduction rates following mandibular advancement, the larger the improvement in REI due to OA. This indicates that the rate of reduction in respiratory resistance is useful for predicting the efficacy of OA therapy. This predictive measure is clinically straightforward, and has good sensitivity and specificity. Although the previous studies have reported that there was a positive correlation between AHI and respiratory resistance [32], and that respiratory resistance was decreased by advancing the mandible [33], the utility of respiratory resistance in the prediction of OA treatment responses has not been shown. The results of this study are the first to demonstrate the clinical utility of respiratory resistance for the prediction of OA treatment response in OSA patients using quantitative methodology. Though various types of OAs are available, OAs are broadly classified into two types: Mono-block and Bi-block types [34]. In Japan, where OA therapy is provided under the National Health Insurance, mono-block OA types are commonly used to balance the costs and the technical simplicity of the treatment procedure. Therefore, a mono-block OA was used in this study.

In this study, there was no significant difference in the R20 at baseline between the responders and non-responders. Similarly, no significant differences were seen between the measurements of the pharyngeal airway spaces and maxillofacial morphology at baseline. Thus, the OA treatment response could not be predicted by the position of the mandible at rest, because there was no difference between anatomical characteristics and the respiratory resistance values between the two groups at baseline. However, despite the fact that the mandibular positions were sufficiently adjusted with the OA in all subjects, the R20 reduction rate according to the mandibular advancement was higher in the responders compared to non-responders. Furthermore, we performed a regression analysis including the conventional predictors (sex, age, BMI, and MR-H) of OA treatment responses in addition to R20. Only the R20 reduction rate was a predictor of OA treatment responses, and no other factors were determined. These results suggested that the reduction rate of respiratory resistance via mandibular advancement was highly correlated with OA treatment effects, and a larger reduction in respiratory resistance was shown to lead to the higher efficacy of OA.

A cephalometric analysis showed that in the responders, the superior/middle/inferior regions of the pharyngeal airway space were all widened by advancing the mandible, with the upper airway widening in a particularly large amount. In the non-responders, widening of the pharyngeal regions with mandibular advancement was not seen. Thus, the reduction of respiratory resistance via mandibular advancement can be expected to result from the widening of the pharyngeal region. In the endoscopy, Sasao et al. [35] reported that there were two types of morphological changes seen in responders to OA therapy: “all-round-type,” which is a circumferential widening in the antero-posterior/lateral directions via mandibular advancement, and a “lateral dominant type,” which is a widening primarily in the lateral direction. Among the responders in this study, some showed poor widening of the pharynx on the cephalogram, despite the large reductions in respiratory resistance by advancing the mandible. Since the cephalometry is capable of observing only morphological changes in the antero-posterior direction, the corresponding subjects presumably belonged to the lateral dominant-type, in which widening occurred in the lateral direction.

The reported conventional testing methods that predict the efficacy of OA therapy include cephalometry [36], computed tomography (CT) [37], magnetic resonance imaging (MRI) [38], and endoscopy [39]. Concerning the prediction of the efficacy of OA therapy using cephalometry, a shorter soft palate, shorter distance between the soft palate and the posterior pharyngeal wall, larger ANB angle, and smaller SNB angle have all been shown to lead to a higher OA efficacy [36]. Similar to in this study, the efficacy of OA therapy has been shown to be high in CT [37], MRI [38], and endoscopy [39] studies, in which a widening of the upper airway via mandibular advancement was seen. However, although these testing methods are useful to predict the efficacy of OA therapy, they require a special testing room and have several issues, such as exposure at the time of imaging/testing, invasion, and a high cost. The protocol that was used to predict the efficacy of OA using respiratory resistance in this study is a simple, non-invasive, low-cost method, because it can be measured by simply purchasing a test device and wearing a nasal mask. Therefore, this method is considered beneficial for all physicians, dentists, and patients involved with OA therapy.

This study has a number of limitations. In this sleep test, electroencephalography was not conducted, and it was not possible to measure the exact sleep time. Therefore, the REI calculated from the test results may be underestimated. In addition, this test cannot distinguish between obstructive and central events. The study included a small sample size, and the number of subjects with varying OSA severities was uneven. In the present study (*n* = 27), the number of severe cases was small (mild [*n* = 10], moderate [*n* = 16], and severe [*n* = 1]), and the subjects were unevenly distributed compared to the typical distributions of patients with OSA. By increasing the sample size, an analysis in patients with severe OSA needs to be conducted in the future. Another potential limitation is the relatively lower BMI than other studies. The World Health Organization Expert Consultation has reported that for many Asian populations, trigger points for public health action were identified as ≧ 23 kg·m^2^, because Asians generally have a higher percentage of body fat than Caucasians of the same BMI [40]. Additionally, the Committee of Japan Society for the Study of Obesity reported that the criteria for obesity disease for Japanese was defined as a BMI ≧ 25 kg·m^2^ [41]. Therefore, our Japanese sample (mean BMI 25.4 kg·m^2^) was considered obese for their standard. To better understand if this study is valid in a Caucasian obese population, further trials with the same methodology are still required. Additionally, there were some cases whose REI reduction rates were not high enough, despite a large reduction in their respiratory resistance via mandibular advancement. Conversely, there were also cases whose REI reduction rates were high, despite small reductions in respiratory resistance. Since we performed comparisons in order to analyze the changes in respiratory resistance while subjects were awake, and measured their respiratory conditions during sleep in the present examination, the differences in environments during the awake state and during sleep may have affected the results. Although the muscles surrounding the pharynx relax during sleep, and the airway is thought to become narrower than while awake, how findings during wakefulness are reflected in the efficacy of OA treatment during sleep remains unclear, and further research is needed. The utility of R20 as a predictor of the success rate of treatment to CPAP was not evaluated in this study; this is necessary to evaluate in a further study. In this study, we prepared a titrated OA and measured the respiratory resistance with and without OA for 2 days. For clinical application, the methods used in this study need to be simpler. As a future prospect, the change in respiratory resistance by advancing the mandibulae will be measured using a George gauge before OA treatment. The response of OA treatment may be predicted by evaluating the change in 1-day measurement and using the cut-off value calculated in this study.

Our study results can be considered to be useful for both physicians seeking OA therapy and dentists responsible for OA therapy. In the current clinical setting, due to difficulty in preoperatively predicting the efficacy of OA therapy, preparing/wearing/adjusting an OA and undergoing a sleep test for the evaluation of efficacy require time and labor. Consequently, when an expected result is not obtained, disadvantages for the patient/dentist/physician are significant, which results in a loss of reliability of OA therapy. In order to avoid such a situation, if suitability of OA therapy can be screened by predicting the efficacy of OA therapy at a medical organization in advance, physicians would then be able to explain the suitability to patients with mild to moderate OSA who desire OA therapy and those who are intolerant to CPAP. In addition, both dentists receiving a request for OA therapy and patients receiving treatment can smoothly proceed with OA therapy based on the prior prediction, and evidence-based medicine (EBM), which is required in modern medicine, can be realized, and accountability can be expected to be fulfilled. Furthermore, although either CPAP or OA therapy is uniformly chosen using the AHI of sleep testing in the current OSA treatment, tailor-made medicine, which determines diagnosis and optimal therapy depending on each case, can be chosen.

## 5. Conclusions

With the aim of evaluating OA indications, respiratory resistance in the waking state, as well as examining how changes in respiratory resistance via mandibular advancement affected the efficacy of the OA treatment, were measured. This study showed that the OA treatment was highly efficient in cases with large reductions in respiratory resistance. In addition, the results suggested that the use of a cut-off value for the reduction rate of respiratory resistance analyzed in this study allowed for the diagnosis and selection of patients in whom OA therapy is effective. However, in order to confirm the reliability and universality of the study results, further analyses need to be performed with a larger sample size, and with uniform conditions in the groups.

## Figures and Tables

**Figure 1 jcm-10-01255-f001:**
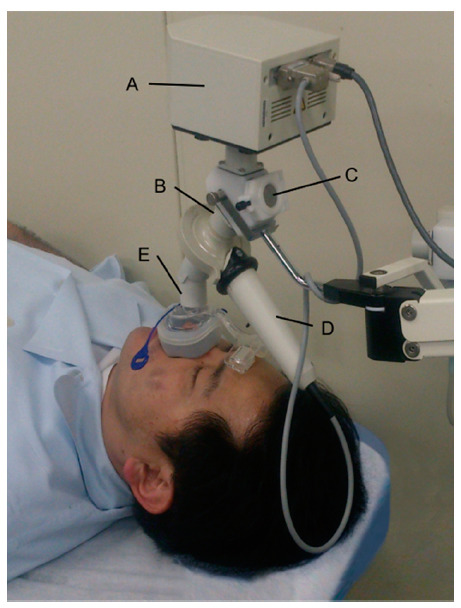
Impulse oscillometry system used in this study: loudspeaker (**A**), screen flap (**B**), Y-adapter (**C**), pnuemochomatograph (**D**), nasal mask (**E**).

**Figure 2 jcm-10-01255-f002:**
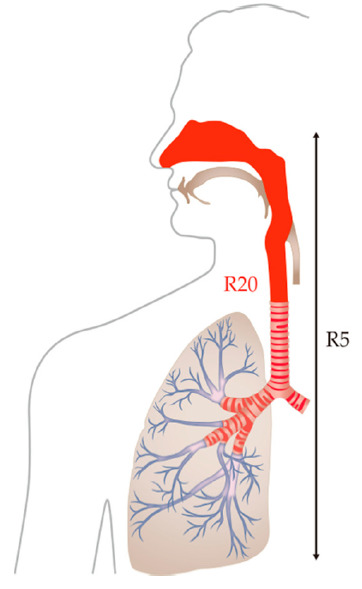
Scheme of Impulse Oscillation System (IOS) values: resistance at 5 Hz (R5); resistance at 20 Hz (R20). R5 and R20 represent the total airway resistance and the resistance of the region from the upper airway through to the central airway, respectively.

**Figure 3 jcm-10-01255-f003:**
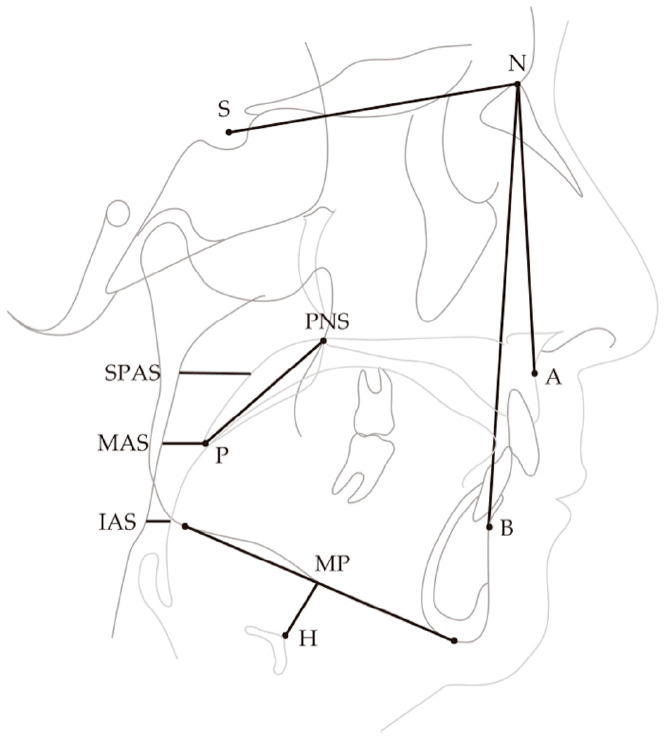
Cephalometric landmarks. The following points were identified on lateral cephalograms. Points: S (sella: the midpoint of the pituitary fossa), N (nasion: the most anterior point on the frontonasal suture), ANS (anterior nasal spine: the tip of the median, sharp bony process of the maxilla at the lower margin of the anterior nasal opening), PNS (posterior nasal spine: the intersection of the continuation of the anterior wall of the pterygopalatine fossa and the floor of the nose, marking the dorsal limit of the maxilla), A (the deepest midline concavity on the anterior maxilla), B (the deepest midline concavity on the mandibular symphysis), P (most inferior tip of the soft palate), H (hyoidale). Planes: MP (mandibular plane according to Steiner: the line through the gonion and gnathion). Linear measurements: PNS-P (linear distance between PNS and P), MP-H (linear distance perpendicular from H to the mandibular plane), SPAS: superior posterior airway space (width of airway behind soft palate along parallel line to gonion (Go)-B line), MAS: middle airway space (width of airway along parallel line to Go-B line through P), IAS: inferior airway space (width of airway along Go-B line).

**Figure 4 jcm-10-01255-f004:**
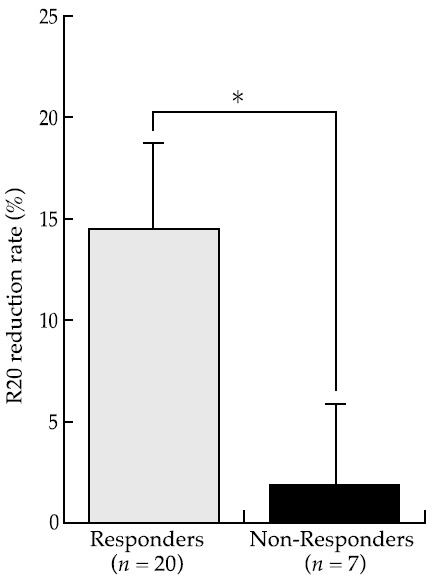
Comparison of responders and non-responders in terms of R20 improvement rates (* *p* < 0.05).

**Figure 5 jcm-10-01255-f005:**
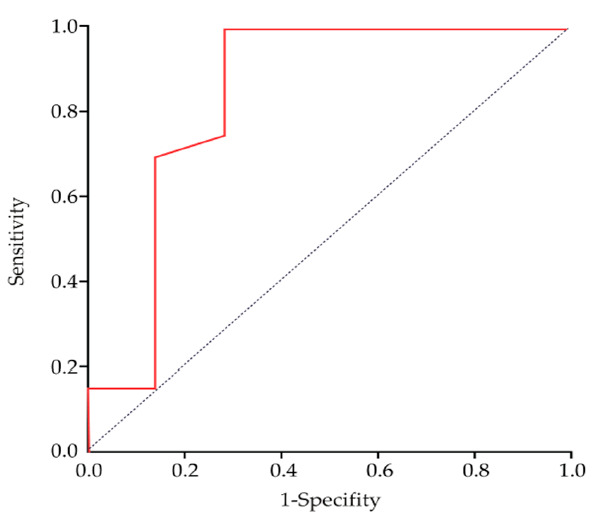
Receiver operating characteristic (ROC) curves of the R20 improvement rates.

**Table 1 jcm-10-01255-t001:** Patient characteristics of the responders and non-responders receiving oral appliance (OA) therapy.

	All (*n* = 27)	Responders (*n* = 20)	Non-Responders (*n* = 7)	*p*-Value ^a^	ES
Sex (%men)	81.5 (22/27)	66.7 (15/20)	100 (7/7)	<0.01	-
Age (years)	65.0 (56.0–70.0)	65.5 (44.3–70.0)	65.0 (58.0–70.0)	0.50	-
BMI (kg/m^2^)	25.1 (22.2–26.8)	26.5 (22.1–28.8)	25.1 (22.2–25.5)	0.31	-
Mallampati class	3.0 (3.0–4.0)	3.0 (3.0–3.8)	4.0 (2.0–4.0)	0.53	-
REI (events/hour)	17.5 ± 6.5	16.4 ± 5.9	20.7 ± 7.1	0.22	0.69
REI severity					
Mild-*n* (%)	9 (33.3)	8 (40.0)	1 (14.3)		
Moderate-*n* (%)	17 (63.0)	11 (55.0)	6 (85.7)		
Severe-*n* (%)	1 (3.7)	1 (5.0)	0 (0.0)		
LowestSpO2 (%)	83.5 ± 5.8	84.1 ± 5.2	81.7 ± 7.1	0.36	0.42
R20 (kPa/L/s)	0.48 ± 0.15	0.49 ± 0.17	0.47 ± 0.11	0.85	0.13
ESS	8.0 (6.0–11.0)	8.0 (7.0–12.5)	6.0 (4.0–8.0)	0.13	-
OAM/MaxM (%)	59.8 ± 10.4	58.8 ± 10.4	62.6 ± 9.8	0.43	0.37
SNA (°)	83.0 ± 4.2	82.4 ± 4.1	84.6 ± 4.2	0.25	0.54
SNB (°)	78.6 ± 4.5	78.1 ± 5.0	79.8 ± 2.3	0.27	0.38
ANB (°)	4.4 ± 2.7	4.3 ± 2.7	4.7 ± 2.5	0.73	0.15
PNS-P (mm)	41.4 ± 4.7	42.0 ± 7.9	39.6 ± 6.5	0.49	0.32
MP-H (mm)	17.3 ± 7.6	16.9 ± 6.6	18.5 ± 9.8	0.64	0.21
SPAS (mm)	4.5 (0.0–9.0)	5.8 (4.7)	0.0 (0.0–9.0)	0.10	0.76
MAS (mm)	13.4 ± 5.4	13.1 ± 5.6	14.2 ± 4.7	0.65	0.20
IAS (mm)	9.7 ± 4.3	9.5 ± 4.0	10.3 ± 5.7	0.67	0.18

Data expressed as mean ± standard deviation (SD), median, and interquartile range (IQR). ^a^ Comparison between responders and non-responders. ES: Effect size (Cohen’s d). BMI: Body Mass Index, REI: Respiratory Event Index, R20: Respiratory resistance at 20 Hz. ESS: Epworth Sleepiness Scale, OAM: Mandibular advancement with appliance. MaxM: Maximum mandibular advancement, SPAS: Superior posterior airway space. MAS: Middle airway space, IAS: Inferior airway space.

**Table 2 jcm-10-01255-t002:** Efficacy of oral appliance therapy in the responders and the non-responders.

	Responders (*n* = 20)			Non-Responders (*n* = 7)		
	w/o OA	with OA	ES	w/o OA	with OA	ES
REI (events/h)	16.4 ± 5.9	5.3 ± 2.1 *	2.51	20.7 ± 7.1	15.5 ± 4.6 *	0.87
Lowest SpO_2_ (%)	84.1 ± 5.2	88.7 ± 4.0 *	1.07	81.7 ± 7.1	83.6 ± 4.9	0.31
R20 (kPa/L/s)	0.49 ± 0.17	0.41 ± 0.13 *	0.5	0.47 ± 0.11	0.46 ± 0.1	0.1
ESS	8.0 (7.0–12.5)	6.0 (4.0–11.5) *	-	6.0 (4.0–8.0)	7.0 (4.0–8.0)	-
SPAS (mm)	5.8 ± 4.7	10.1 ± 6.2 *	0.8	2.4 ± 3.6	4.3 ± 5.8	0.39
MAS (mm)	13.1 ± 5.6	15.9 ± 5.7 *	0.5	14.2 ± 4.7	15.8 ± 4.7	0.34
IAS (mm)	9.5 ± 4.0	11.8 ± 4.6 *	0.53	10.3 ± 5.0	9.3 ± 5.6	0.19

Data expressed as mean ± standard deviation (SD), median, and interquartile range (IQR). * *p* < 0.05. ES: Effect size (Cohen’s d), REI: Respiratory Event Index, R20: Respiratory resistance at 20 Hz, ESS: Epworth Sleepiness Scale, SPAS: Superior posterior airway space, MAS: Middle airway space, IAS: Inferior airway space.

**Table 3 jcm-10-01255-t003:** Logistic regression results for factors associated with oral appliance treatment success.

Variables	*β*	*p*-Value	Odds Ratio	95% CI
Improvement in R20				
0.1% increments	0.16	0.018	1.17	1.03–1.34
1% increments	0.16	0.018	4.95	4.33–5.66
2% increments	0.16	0.018	24.5	21.5–28.0

95% CI: 95% Confidence Interval. R20: Respiratory resistance at 20 Hz.

## Data Availability

The data presented in this study are available on request from the corresponding author.

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
