# Peer review of "Evaluation of Respiratory Resistance as a Predictor for Oral Appliance Treatment Response in Obstructive Sleep Apnea: A Pilot Study"

_jcm, 2021, doi:10.3390/jcm10061255_

Round 1

Reviewer 1 Report

This is a nice paper and is very clearly written. The authors explore the use of impulse oscillometry to predict oral appliance treatment response. They use the R20 measure, which is a measure of predominantly upper airway resistance. The main finding is that the decreases in R20 with mandibular advancement is predictive of treatment response. They are the first to report the use of R20 for prediction of oral appliance efficacy. Methods are sound and the results are clearly reported and discussed. The main limitation of the technique is that it is performed over 3 separate days and requires an already titrated oral appliance. One could argue that there are simpler ways to predict treatment response once the oral appliance has already been titrated and provided to the patient (e.g. home sleep test – even something as simple as a pulse oximeter worn at night). Nonetheless, the method holds promise as it has the potential to be simplified. For example, IOS can be measured in a single day while protruding the jaw manually or with a temporary oral appliance made that day (e.g. “boil-and-bite” oral appliance).  

Abstract

  • In abstract you state: “R20 reduction rate was predictive for OA treatment responses” – please list some results of logistic regression model that lead to the conclusion that R20 is predictive.

Introduction

  • Line 43-44: Please include a reference for high compliance of oral appliance treatment (Diejtens et al. Chest 2013; Vanderveke et al. Thorax 2013)

Methods

  • AHI cutoff for inclusion of 5 events/hr is quite low. Is it justified that a patient that goes from AHI 5 to 3 /hr is considered a non-responder? What is your justification for including patients with such low AHI? I understand it might be because of the small sample, in which case I would suggest that in the larger study you set the cutoff at 10 /hr minimum.
  • Line 91: why do you specify that expiratory flow was measured with nasal cannula. Both inspiration and expiration are measured by nasal cannula.
  • Paragraph starting on line 94: it is not clear what “testing” you are referring to. The sleep test or IOS?
  • Line 116: “During follow-up” – what follow-up? Is this the follow-up following the provision of the oral appliance? If so, at roughly what time frame was this follow-up? Days, weeks, months?
  • Can you include REI in the logistic regression model? I think it is important to show that it is not just the non-severe patients that are responding. I don’t think it will change anything since responders don’t have a significantly lower baseline AHI.

Results

  • A figure showing the subject level reduction in REI from baseline to treatment in responder and non-responder groups, indicating patients correctly predicted, would be very helpful. I am mostly curious how the mild cases were classified in terms of response.

Discussion

  • In general, I think the main limitation of the method is that you measured airway resistance over 3 days and required a titrated oral appliance. Requiring patients come to clinic 3 times is quite cumbersome. Compounded by the fact that the OA has already been titrated, you could argue a simple home sleep test (e.g. get ODI from pulse oximeter) is easier. I think there are ways around this (measure all in one day and use a temporary appliance), but this needs to be tested. I would include a brief discussion of this in the limitations.
  •  

Author Response

Response to Reviewer 1 Comments

We are pleased to resubmit for publication the revised version of No. jcm-1092382 entitled “Evaluation of Respiratory Resistance as a Predictor for Oral Appliance Treatment Response in Obstructive Sleep Apnea: A Pilot Study". We appreciate the constructive criticisms of the Reviewers. We have addressed each of their concerns as outlined below.

Point 1: In abstract you state: “R20 reduction rate was predictive for OA treatment responses” – please list some results of logistic regression model that lead to the conclusion that R20 is predictive.

Response 1: We agree with the reviewer and the following information was added to the manuscript (lines 31-33):

"A logistic multiple regression analysis showed that the R20 reduction rate was predictive for OA treatment responses (2% incremental odds ratio (OR), 13.4; 95% CI, 11.8-15.3; p = 0.018)."

Point 2: Line 43-44: Please include a reference for high compliance of oral appliance treatment (Diejtens et al. Chest 2013; Vanderveke et al. Thorax 2013)

Response 2: We agree with the reviewer. We added two references (Diejtens et al. Chest 2013; Vanderveken et al. Thorax 2013).

Point 3: AHI cutoff for inclusion of 5 events/hr is quite low. Is it justified that a patient that goes from AHI 5 to 3 /hr is considered a non-responder? What is your justification for including patients with such low AHI? I understand it might be because of the small sample, in which case I would suggest that in the larger study you set the cutoff at 10 /hr minimum.

Response 3: The sample size in this study was small. Therefore, we will set the cutoff value of 10 events/hr in further studies.

Point 4: Line 91: why do you specify that expiratory flow was measured with nasal cannula. Both inspiration and expiration are measured by nasal cannula.

Response 4: We agree with the reviewer and we changed the manuscript as follows (lines 93-95):

“Expiratory and inspiratory flow was measured using a nasal cannula, and respiratory conditions during sleep were recorded.”

Point 5: Paragraph starting on line 94: it is not clear what “testing” you are referring to. The sleep test or IOS?

Response 5: We agree with the reviewer and we changed the manuscript as follows (lines 96-97):

“The Sleep test was performed for three consecutive nights, and mean values were used as representative values.”

Point 6: Line 116: “During follow-up” – what follow-up? Is this the follow-up following the provision of the oral appliance? If so, at roughly what time frame was this follow-up? Days, weeks, months?

Response 6: We agree with the reviewer. We changed the manuscript as follows (lines 118-119):

“During follow-up of 2 months after the OA provision, the changes in OSA symptoms and the degrees of side effects that were associated with OA use were evaluated.”

Point 7: Can you include REI in the logistic regression model? I think it is important to show that it is not just the non-severe patients that are responding. I don’t think it will change anything since responders don’t have a significantly lower baseline AHI.

Response 7: We agree with the reviewer. We included REI in the objective variables and performed logistic regression analysis again (Table 3).

Point 8: A figure showing the subject level reduction in REI from baseline to treatment in responder and non-responder groups, indicating patients correctly predicted, would be very helpful. I am mostly curious how the mild cases were classified in terms of response.

Response 8: We agree with the reviewer. We added the OSA severity in patient characteristics of the responders and non-responders (Table 1).

Point 9: In general, I think the main limitation of the method is that you measured airway resistance over 3 days and required a titrated oral appliance. Requiring patients come to clinic 3 times is quite cumbersome. Compounded by the fact that the OA has already been titrated, you could argue a simple home sleep test (e.g. get ODI from pulse oximeter) is easier. I think there are ways around this (measure all in one day and use a temporary appliance), but this needs to be tested. I would include a brief discussion of this in the limitations.

Response 9: In this study, we prepared a titrated OA and measured the respiratory resistance for 2 days with and without OA. Therefore, it is not a measurement of respiratory resistance for 3 days. The following information was added to the limitations (lines 364-369):

“In this study, we prepared a titrated OA and measured the respiratory resistance for 2-days with and without OA. For future clinical application, the method used in this study needs to be a simpler method. As a future prospect, the change in respiratory resistance by advancing the mandibulae will be measured using a George gauge before OA treatment. The effect of OA treatment can be predicted by evaluating the change in 1-day measurement and using the cut-off value calculated in this study.”

Reviewer 2 Report

The work is original and innovative. It is well exposed and the topic is very interesting.

This pilot study enrolled a small sample of sleep apnoea patients, with a prevalent low-moderate grade, to measure respiratory resistance using a non-invasive method called Impulse Oscillation System in the waking state. One of the aim was to evaluate the respiratory resistance via mandibular advancement, examining how oral appliance could modify sleep apnoea.

To the best of my knowledge, in literature, this system was poorly used and never specifically for oral appliance. It may be the first study to combine the craniometrics evaluation with a respiratory resistance measure.

Their results showed that Oral appliance treatment highly reduce respiratory resistance and that this test may be useful in the selection of patients.

Therefore, I think the work is very interesting and relevant, especially because it proposed a non-invasive method, easy to perform, that could be conducted in a short period.

Finally, the paper is well written, all paragraphs are well expressed and complete. Authors totally describe methods and results, showing study limitations and potentialities. The text results easy to read and comprehend. The conclusions are consistent, in accordance with the description proposed.

In my opinion, all possible doubts that emerged are solved during the reading.

Author Response

Response to Reviewer 2 Comments

We are pleased to resubmit for publication the revised version of No. jcm-1092382 entitled “Evaluation of Respiratory Resistance as a Predictor for Oral Appliance Treatment Response in Obstructive Sleep Apnea: A Pilot Study". We appreciate the constructive criticisms of the Reviewers.

Reviewer 3 Report

The manuscript entitled „Evaluation of Respiratory Resistance as a Predictor for Oral Appliance Treatment Response in Obstructive Sleep Apnea: A Pilot Study” reports on the utility of respiratory resistance as a predictor of oral appliance response in obstructive sleep apnea.

In the methods section, there is no information about the number of patients, who were recruited for the study (and how many were excluded). This information is necessary to assess how selective was the inclusion process.

Please add the number of patients excluded from the study due to inconsistent data.

Please explain why “Those allergic to steroids and vasoconstrictors.” Was considered as exclusion criteria.

In the statistical analysis paragraph please add if the distribution of data was performed (and if so, what test was used). All data are presented as mean with standard deviation this suggests that all the data had a normal distribution, which is highly unlikely with such a small study group. For non-normal distribution, data should be presented as median and IQR. Furthermore, the authors state that Mann-Whitney U (misspelled in the manuscript, missing n in Whitney), which is based on the presentation of data is wrongly used as it is dedicated to nonnormal data. Therefore, it is impossible to assess the results in the full, as the p-value can differ from the stated one when a different (proper) test will be used.

While assessing the cut-off point ROC curve authors use the Youden index. It could be more interesting and useful, to find a point, in which sensitivity is the highest to use it to exclude patients, who will not react to the OA treatment.

In the discussion, the authors state that there were no differences in the “the measurements of the pharyngeal airway spaces and maxillofacial morphology at baseline” without providing such data in results. Fe. Malampati should be used and compare to R20, to provide some reference to obtained data.

Lack of comparison of the effectiveness of R20 as a predictor of the success rate of treatment to CPAP is a weak point of the study. Similarly, only one individual with a severe form of OSA (the information we are presented with as late as discussion) is a weekend of the study as it misrepresents this group of patients.

While in theory, the study suggests a possible form of predicting the success rate of OA treatment, in practice, with a sensitivity of 0.72, almost 30% of individuals would be classified as responders and would undergo the procedure without success. In a case where an individual does not have any indications for surgery as a treatment, losing weight, has non-positional OSA, does not tolerate/agree to CPAP treatment, OA is the only option regardless of prediction of response to OA.

Page 1, line 45/page 2, line 46: while stating that OA improves cardiovascular diseases, please add that it refers to comorbid CVDs.

Page 2, line 46/47: while stating the success rate of OA please add, to which group of OSA patients it refers to (severity, etc.).

Page 2, line 47/48: the statement“its efficacy is known to be high in some patients and low in others” is unclear and should be rephrased; fe. Efficacy differs is based on several parameters. 

Page 2, line 69: sentence “(…) treatment of bronchial asthma and chronic obstructive pulmonary disease (COPD). „ should be followed by an appropriate reference.

Author Response

Response to Reviewer 3 Comments

We are pleased to resubmit for publication the revised version of No. jcm-1092382 entitled “Evaluation of Respiratory Resistance as a Predictor for Oral Appliance Treatment Response in Obstructive Sleep Apnea: A Pilot Study". We appreciate the constructive criticisms of the Reviewers. We have addressed each of their concerns as outlined below.

Point 1: In the methods section, there is no information about the number of patients, who were recruited for the study (and how many were excluded). This information is necessary to assess how selective was the inclusion process. Please add the number of patients excluded from the study due to inconsistent data.

Response 1: We agree with the reviewer. We changed the manuscript as follows (lines 213-214):

“Thirty-five patients were recruited into this study, and 8 OSA patients refused to participate in the study. 27 OSA patients completed the full study protocol.”

Point 2: Please explain why “Those allergic to steroids and vasoconstrictors.” Was considered as exclusion criteria.

Response 2: Because it is a contraindication to a vasoconstrictor nasal drop (cor-tyzine nasal solution; Tetrahydrozoline hydrochloride-prednisolone) used in this study, we set it as an exclusion criterion.

Point 3: In the statistical analysis paragraph please add if the distribution of data was performed (and if so, what test was used). All data are presented as mean with standard deviation this suggests that all the data had a normal distribution, which is highly unlikely with such a small study group. For non-normal distribution, data should be presented as median and IQR. Furthermore, the authors state that Mann-Whitney U (misspelled in the manuscript, missing n in Whitney), which is based on the presentation of data is wrongly used as it is dedicated to nonnormal data. Therefore, it is impossible to assess the results in the full, as the p-value can differ from the stated one when a different (proper) test will be used.

Response 3: We agree with the reviewer. We performed the statistical analysis again (lines 190-197):

“Continuous variables were described as mean ± SD for variables with a normal distribution and median (interquartile range) for variables with a non-normal distribution. Normality of distribution was assessed using the Shapiro-Wilk test. The comparisons between groups were made using student’s t-test for variables with a normal distribution, and Mann-Whitney U test for variables with a non-normal distribution, and Fisher’s exact test for categorical variables. The within-subject comparisons in each group were made using pared t-test for variables with a normal distribution.”

Point 4: While assessing the cut-off point ROC curve authors use the Youden index. It could be more interesting and useful, to find a point, in which sensitivity is the highest to use it to exclude patients, who will not react to the OA treatment.

Response 4: We agree with the reviewer. We made a wrong analysis of the ROC curve. We corrected the content and performed statistical analysis again (lines 269-271):

“The calculated cut-off value of the R20 reduction rate from the Youden index was 8.6% (AUC: 0.839, sensitivity: 0.80, specificity: 0.86) (Figure 5).”

Point 5: In the discussion, the authors state that there were no differences in the “the measurements of the pharyngeal airway spaces and maxillofacial morphology at baseline” without providing such data in results. Fe. Mallampati should be used and compare to R20, to provide some reference to obtained data.

Response 5: We agree with the reviewer. We added the comparison results of Mallampati classification in patient characteristics of the responders and non-responders (Table 1).

Point 6: Lack of comparison of the effectiveness of R20 as a predictor of the success rate of treatment to CPAP is a weak point of the study. Similarly, only one individual with a severe form of OSA (the information we are presented with as late as discussion) is a weekend of the study as it misrepresents this group of patients.

Response 6: We agree with the reviewer and the following information was added to the limitations (lines 362-364):

“The utility of R20 as a predictor of the success rate of treatment to CPAP was not evaluated in this study. It will be necessary to evaluate in further study.”

Point 7: While in theory, the study suggests a possible form of predicting the success rate of OA treatment, in practice, with a sensitivity of 0.72, almost 30% of individuals would be classified as responders and would undergo the procedure without success. In a case where an individual does not have any indications for surgery as a treatment, losing weight, has non-positional OSA, does not tolerate/agree to CPAP treatment, OA is the only option regardless of prediction of response to OA.

Response 7: We agree with the reviewer. Although either CPAP or OA therapy is uniformly chosen using AHI of sleep testing in the current OSA treatment, tailor-made medicine, which determines diagnosis and optimal therapy depending on each case, can be chosen by the results in this study.

Point 8: Page 1, line 45/page 2, line 46: while stating that OA improves cardiovascular diseases, please add that it refers to comorbid CVDs.

Response 8: We agree with the reviewer. The following information was added to the manuscript (lines 43-47):

“Although its efficacy is inferior to that of continuous positive airway pressure (CPAP) therapy, OA therapy has high compliance [6,7] and not only improves apnea, hypopnea, and symptoms of OSA, but has recently also been shown to be effective on the cardiovascular comorbidity in OSA, such as hypertension and arrhythmia [8].”

Point 9: Page 2, line 46/47: while stating the success rate of OA please add, to which group of OSA patients it refers to (severity, etc.).

Response 9: We agree with the reviewer. The following information was added to the manuscript (lines 47-48):

“The success rate of OA therapy in patients with mild–severe OSA was reported to be 35–64% in a recent review [9].”

Point 10: Page 2, line 47/48: the statement“its efficacy is known to be high in some patients and low in others” is unclear and should be rephrased; fe. Efficacy differs is based on several parameters. 

Response 10: We agree with the reviewer. The following information was added to the manuscript (lines 48-50):

“individual variability in response to OA treatment represented a significant clinical challenge, as implementing therapy [10].”

Point 11: Page 2, line 69: sentence “(…) treatment of bronchial asthma and chronic obstructive pulmonary disease (COPD). „ should be followed by an appropriate reference.

Response 11: We agree with the reviewer. We added the reference (Ohishi, J et al. BMJ Open 2011).